# NEOL: Reward-Gated Online Plasticity for Scalable NeuroEvolution

## Abstract

NeuroEvolution of Augmenting Topologies (NEAT) excels at discovering neural architectures and weights for control tasks (Stanley & Miikkulainen, 2002a). However, direct-encoding forces evolution to discover each connection strength individually; in high-dimensional weight spaces, this yields weak credit assignment and poor scaling on large continuous-control problems (Stanley et al., 2009; Peng et al., 2018). We propose NeuroEvolutionary Online Learning (NEOL), which decouples learning signals: the outer loop uses NEAT for topology search, while an inner, reward-modulated local plasticity rule (Hebbian, Oja, or BCM (Hebb, 1949; Oja, 1982; Bienenstock et al., 1982)) adapts synaptic weights online within episodes. Under fixed interaction budgets and multiple seeds across four standard control benchmarks spanning discrete and continuous action spaces, NEOL achieves higher final returns, tighter variability, and better sample efficiency than pure NEAT; gains are most pronounced in continuous control. These improvements are statistically significant (Wilcoxon rank-sum tests), and ablations indicate that benefits persist even when standard genetic weight mutation is reduced or disabled, evidencing a division of labour between structural evolution and online synaptic credit assignment. A simple, gradient-free separation of topology search and reward-gated online plasticity reliably boosts performance and robustness, offering a practical template for linking neuroevolution with online learning and a scalable path toward more adaptive neuroevolutionary agents.

## 1 Introduction

NeuroEvolution employs evolutionary operators, rather than gradient descent, to optimise neural network architectures and parameters (Miikkulainen, 2025; Miikkulainen et al., 2024; Stanley et al., 2019; Khan et al., 2010; Yao, 1999; Angeline et al., 1994). It has been applied to biologically inspired models for lifelong learning (Kudithipudi et al., 2022), reinforcement learning (RL) problems (Xue et al., 2024; Chalumeau et al., 2023; Co-Reyes et al., 2021; Khadka & Tumer, 2018; Stanley & Miikkulainen, 2002b), and even domains such as optimising land-use planning policies to reduce carbon emissions (Young et al., 2025). Among NeuroEvolution methods, NeuroEvolution of Augmenting Topologies (NEAT) is a landmark approach: it jointly discovers network topology and weights and is particularly effective on tasks that require structural innovation (Stanley & Miikkulainen, 2002a; 2004).

However, relying solely on evolution for weight optimisation poses a major challenge in high-dimensional spaces: mutation-based perturbations provide weak credit assignment, leading to poor sample efficiency and a tendency to converge to local optima in complex continuous control (Stanley et al., 2009). Moreover, networks evolved in a standard offline NeuroEvolution pipeline, where a population is evolved on a task and then fixed for deployment, often struggle with real-time interaction and sequential decision making (Agogino et al., 2000; Bellman, 1966; Sutton et al., 1998). These limitations have motivated hybrid approaches that combine evolutionary structure search with more direct procedures for weight adaptation (Peng et al., 2018).

Several such hybrids have been explored. HyperNEAT (Stanley et al., 2009) exploits geometric regularities by mapping task structure onto network topology, shifting difficulty from dimensionality to problem structure (Stanley et al., 2009). Nevertheless, subsequent studies report that HyperNEAT can underperform NEAT on some large-scale problems, including Atari games, and struggle when

the state–action mapping is highly discontinuous (the "fracture" issue) (Hausknecht et al., 2014; Kohl & Miikkulainen, 2009). Other works combine NEAT with value-based or policy-gradient RL. For example, Whiteson et al. (2005) integrated Q-learning with NEAT, but demonstrated results only on a single control domain (Robot Auto Racing Simulator), leaving scalability uncertain. Peng et al. (2018) proposed NEAT with Policy Gradient Search (NEAT-PGS), where RL is typically used to pre-train policy networks, NEAT evolves a feature network, and the policy is then further trained given the evolved features.

These observations raise a natural question: can we use NeuroEvolution to directly train policy networks for game-playing tasks, avoiding long and potentially unstable training chains while still achieving robust performance? Early work by Agogino et al. (2000) explored online evolution of each agent's policy using a fixed 8-5-2 feedforward architecture (single hidden layer, no recurrence, no evolving topology). While this design is lightweight and fast for online evolution, it limits applicability to more general and higher-dimensional tasks, and its evaluation focused on a simplified "mini-Warcraft II" benchmark, where agents with $8$ sensors navigate a small 2D map to reach gold mines while avoiding a single enemy.

Since then, advances in meta learning, deep learning, reinforcement learning, and computational resources have made it feasible for NeuroEvolution to evolve far more complex network topologies and to scale neural networks substantially (Stanley et al., 2019). At the same time, this scalability amplifies a long-standing weakness: mutation-driven local search over high-dimensional weight spaces provides poor credit assignment and is sample inefficient. A natural response is to separate concerns across time horizon: let evolution discover structure over generations, and let online interaction adapt weights within episodes. In particular, if we decompose policy learning into (i) evolutionary search over topology and (ii) online weight adaptation from interaction feedback, we can exploit structural exploration while using reward signals to provide immediate, local credit assignment, improving optimisation and sample efficiency in NeuroEvolution. However, instantiating this separation in a principled and effective way is non-trivial. Therefore, this paper aims to answer:

(1) How should we decompose policy learning, specifically for NEAT, so that direct online training becomes more robust in terms of rewards and more sample efficient?

(2) How should we design the online learning mechanism and propagate reward signals to weights so that interaction feedback is incorporated effectively during training?

**Contribution.** This paper addresses open challenges in robustness and sample efficiency for policy learning in NEAT by decomposing training into weight updates and topology updates and by propagating reward signals. We develop a NeuroEvolution Online Learning (NEOL) framework. To the best of our knowledge, although many RL and other methods have been proposed in the context of NeuroEvolution (Co-Reyes et al., 2021; Miconi et al., 2019; Stanley et al., 2009; Whiteson et al., 2005; Agogino et al., 2000), this is the first use of online learning via synaptic plasticity within NEAT training. We provide extensive experimental evidence that NEOL methods are competitive with standard NEAT in cumulative rewards over the time horizon and in sample efficiency. Specifically, using environments taken from the RL literature, we compare three online learning methods based on synaptic plasticity, including the Hebbian rule, Oja's rule, and the BCM rule. These findings highlight the potential of online learning via synaptic plasticity for NeuroEvolution in interactive RL environments and clarify the role of online learning in effective game-playing.

## 2 PRELIMINARIES AND BACKGROUND

First, we provide a formal formulation of sequential decision making by using the Markov Decision Process (MDP). Given an MDP defined by a tuple $\langle \mathcal{S}, \mathcal{A}, \mathcal{P}, \mathcal{R}, \gamma, T \rangle$ where $\mathcal{S}$ is the state space, $\mathcal{A}$ is the action space, $\gamma \in [0, 1]$ is the discount factor, $\mathcal{R} : \mathcal{S} \times \mathcal{A} \to \mathbb{R}$ is the reward function and $\mathcal{P} : \mathcal{S} \times \mathcal{A} \to \mathcal{S}$ is the transission function. In this paper, we consider an online RL setting where the agent can interact with the environment repeatedly until a certain time horizon $T$ by using a policy $\pi : \mathcal{S} \to \mathcal{A}$. Such an agent's policy is usually represented by a neural net. Then, the goal of the entire learning process is to find an optimal policy $\pi^*$ such that it can maximise the expected

discounted long-term rewards:

$$\pi^* \in \arg\max_\pi \mathrm{E}_\pi \left( \sum_{t=0}^{T} \gamma^t \mathcal{R}\left(s_t, a_t\right) \mid s_0, a_0 \right).$$

As a special setting for NeuroEvolution, this paper directly refers to the cumulative reward as the fitness of the policy at time horizon $T$, i.e., $f(\pi, T) := \sum_{t=0}^{T} \gamma^t \mathcal{R}\left(s_t, a_t\right)$, $s_t, s_0 \in \mathcal{S}$, $a_t, a_0 \in \mathcal{A}$.

**Learning in Games.** Learning in games is a fundamental challenge in machine learning and artificial intelligence, with wide-ranging applications from board games to robust optimisation and agents' game-playing (Silver et al., 2016; Schrittwieser et al., 2020). In this context, games broadly refer to strategic interactions among players and environments, which may be adversarial or collaborative. A particularly rich setting emerges in reinforcement learning problems, including classical control tasks and gridworld tasks, where the goal is to succeed against strategic environments, train effective policies, and achieve high cumulative rewards for agents (Co-Reyes et al., 2021; Khadka & Tumer, 2018; Salimans et al., 2017; Moriarty et al., 1999; Sutton et al., 1998).

**Online Learning via Synaptic Plasticity.** Online learning employs a sequential protocol in which the learner repeatedly predicts, receives feedback, and immediately updates its hypothesis, aiming to minimise cumulative regret even under non-stationary or adversarial scenarios (Shalev-Shwartz, 2011). Unlike batch learning, updates are performed incrementally on a per-example basis without revisiting the entire data set, which naturally suits streaming data and continual adaptation (Shalev-Shwartz, 2011). Mammalian brains support effective online learning, adapting as experience unfolds. A core mechanism underpinning this capability is synaptic plasticity, which denotes activity-dependent changes in synaptic efficacy and has long been regarded as a cellular substrate of learning and memory (the plasticity–memory hypothesis) (Martin et al., 2000; Takeuchi et al., 2014). More precisely, long-term potentiation and long-term depression (LTP/LTD) are widely observed in mammalian excitatory synapses and support experience-dependent circuit remodelling and behavioural learning (Malenka & Bear, 2004; Nicoll, 2017). Spike-timing dependent plasticity (STDP) refines Hebbian learning by making synaptic changes depend on the precise millisecond timing between pre- and postsynaptic spikes, and has been demonstrated across species and brain areas (Caporale & Dan, 2008; Sjöström & Gerstner, 2010; Feldman, 2012).

At the behavioural time scale, many forms of plasticity are modulated by a third factor (for example, neuromodulators such as dopamine or acetylcholine), yielding three-factor learning rules that enable reward-gated and behaviourally relevant credit assignment (Frémaux & Gerstner, 2016; Gerstner et al., 2018; Frémaux et al., 2010). Coupling a global reward signal with local activity correlations enables online reinforcement learning and distal credit assignment without backpropagating gradients through time (Seung, 2003; Florian, 2007; Xie & Seung, 2004). In machine learning, differentiable formulations of plasticity and neuromodulation have improved fast adaptation in few-shot and continual settings, demonstrating practical benefits of embedding synapse-level online learning in artificial neural networks (Miconi et al., 2018; 2019).

Let $x$ denote presynaptic activity, $y$ postsynaptic activity, $w$ a synaptic weight, and $\eta > 0$ a learning rate.

HEBBIAN RULE. Hebb's postulate states that the connection between two neurons strengthens when they are coactive ("cells that fire together wire together"), yielding a simple local, correlation-based weight update (Hebb, 1949; Caporale & Dan, 2008):

$$\Delta w = \eta\, x\, y. \tag{1}$$

While biologically plausible and fully local, pure Hebbian updates are unstable without additional constraints, as weights can diverge (Caporale & Dan, 2008).

OJA'S RULE. Oja introduced a Hebbian update with an implicit normalisation term that prevents divergence and aligns the weight vector with the first principal component under stationary inputs (Oja, 1982):

$$\Delta w = \eta\, y\, (x - y\, w). \tag{2}$$

This modification stabilises learning and endows the single neuron with a principled PCA interpretation (Oja, 1982).

BCM RULE. The Bienenstock–Cooper–Munro (BCM) theory proposes a sliding, activity-dependent threshold that separates LTD from LTP and supports the emergence of selectivity while maintaining homeostasis (Bienenstock et al., 1982):

$$\Delta w \;=\; \eta\, y\, (y - \theta)\, x, \qquad \dot{\theta} \;=\; \alpha\left(y^2 - \theta\right), \tag{3}$$

where $\theta$ is a slow-moving threshold tracking recent activity and $\alpha > 0$ controls its timescale (Bienenstock et al., 1982).

**Bridging Evolutionary Computation and Online Learning.** The idea of combining population-based search with lifelong learning is a foundational aim for achieving adaptive intelligence (Schmidhuber, 1987; Holland, 1992; Miikkulainen, 2025). However, early attempts to evolve synaptic plasticity rules directly within NeuroEvolution faced a key limitation: expanding the genetic search space often hindered learning rather than helping (Stanley et al., 2003). A notable advance was neuromodulation, in which reward-like signals gate local Hebbian updates. This approach proved highly effective, enabling networks to solve dynamic, reward-based tasks that were intractable for both fixed-weight and non-modulated plastic networks (Soltoggio et al., 2008). This success helped establish a powerful paradigm: an outer loop of evolution that designs an inner loop online learner (Soltoggio et al., 2018). This two-timescale approach has since been explored from multiple angles. To address challenges such as deceptive search landscapes and scalability, diversity-driven methods such as novelty search (Lehman & Stanley, 2008; 2011) and indirect encodings such as adaptive HyperNEAT have been developed (Risi, 2012). From a meta optimisation perspective, methods such as Population-Based Training (PBT) have provided practical validation for using asynchronous evolutionary search to supervise and adapt inner learning dynamics online (Jaderberg et al., 2017). More recently, this paradigm has been extended further. Hebbian meta learning has evolved synapse-specific rules that allow agents to adapt rapidly in complex reinforcement learning tasks (Najarro & Risi, 2020). Going further, research has shown that evolution can discover or refine entire RL algorithms from scratch, yielding domain agnostic solutions with strong generalisation (Co-Reyes et al., 2021). In this paper, we provide a systematic empirical study across multiple game benchmarks that contrasts pure NEAT with reward-modulated plasticity NEAT, and compares several online rules (Hebbian, Oja, BCM). Our results show that modulated NEAT consistently outperforms pure NEAT, especially in continuous action spaces.

## 3 NEUROEVOLUTIONARY ONLINE LEARNING

In this section we present the NeuroEvolutionary Online Learning (NEOL) framework. We begin with a high-level overview, then describe the decoupled update strategy for weights and topology in Section 3.1, and finally detail the main algorithm in Section 3.2. Additional components are provided in Appendix A.

### 3.1 DECOUPLING UPDATES FOR WEIGHT AND TOPOLOGY

We maintain a population of network architectures, each encoded as a genome. The flowchart in Fig. 1 shows a generational loop in which evolution and evaluation are interleaved to progressively improve solutions. Unlike standard NEAT, which mutates both topology and weights offline between episodes and evaluates policies with fixed weights during rollouts, NEOL decouples the two update processes. During each individual rollout, synaptic plasticity performs online weight adaptation driven by reward feedback; only after the rollout are topological changes applied by evolutionary variation. This separation places credit assignment for weights on the interaction timescale while reserving structural innovation for the generational timescale, improving sample use and stabilising search (see Section 4).

The process begins with population network structure initialisation (top left, red box in Fig. 1), which creates the initial set of parent networks. The population then enters the main neuroevolution loop (large yellow box). At the start of each generation, the current parents produce offspring through variation (left panel). Variation mutates topology by adding or removing nodes and connections, as is standard in neuroevolution. Each offspring is then evaluated in the individual rollout phase (centre panel). During the rollout, the agent interacts with the environment: the network outputs actions and receives rewards in a closed loop. A key feature of our approach is online weight adaptation,

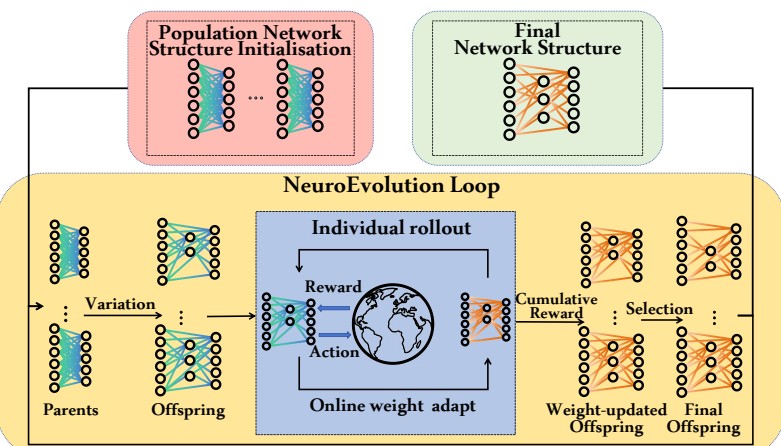

Figure 1: Overview of the NeuroEvolution Online Learning (NEOL) algorithm. The procedure starts by initialising a population of network structures. In each generation, variation operators produce offspring from the current parents. Each offspring is evaluated in an individual rollout, during which its weights adapt online through reward-modulated plasticity (illustrated by the colour shift from blue to orange). The cumulative reward from the rollout is used as the fitness for selection to form the next generation. The best individual found over the population determines the final network structure, including its adapted weights and evolved topology.

where reward-modulated plasticity rules (for example, Hebbian or BCM) update connection weights within the episode. This adaptation is depicted in the figure by connections changing from cool to warm colours. After the rollout, performance is summarised as the cumulative reward, which serves as fitness for selection (right panel). Based on these fitness scores, weight updated offspring are selected to become the parents of the next generation, closing the loop. The best performing individual across all generations is tracked, and its architecture, together with the adapted weights and evolved topology, is reported as the final network structure (top right, green box) when the algorithm terminates.

## 3.2 PROPOSED ALGORITHM

---

**Algorithm 1** NeuroEvolutionary Online Learning (NEOL)

---

Input: Generations: $G \in \mathbb{N}_{>0}$; Population size: $P \in \mathbb{N}_{>0}$; NeuroEvolution config: $\Theta_0$ (Described in Algorithm 3). Parameters for fitness evaluation: Episodes $N \in \mathbb{N}_{>0}$; Max steps $T_{\max} \in \mathbb{N}_{>0}$; Learning rule $\mathcal{L} \in \{\text{Hebb, Oja, BCM}\}$; Plasticity rate $\eta \in \mathbb{R}_{>0}$; Reward scaling $\beta \in \mathbb{R}_{>0}$; Environment env.

Output: Best evolved genome $g^*$.

  1: $\mathcal{P} = \text{INITIALISEPOPULATION}(\Theta_0, P)$
  2: **for** gen $\in \{1, \dots, G\}$ **do**
  3:     **for all** genome $g_i \in \mathcal{P}$ **do**
  4:         $g_i.\text{fitness} = \text{ONLINEROLLOUT}(g_i, \mathcal{L}, \eta, \beta, N, T_{\max}, \text{env})$       $\triangleright$ See Algorithm 2
  5:     $\mathcal{P} = \text{REPRODUCE}(\mathcal{P})$             $\triangleright$ For NEAT: See Algorithm 3 and 4
  6: **return** best genome $g^*$ from final population $\mathcal{P}$

---

Algorithm 1 implements a two-timescale procedure. A population of size $P$ is initialised from $\Theta_0$. In each generation, every genome $g_i$ is evaluated by Algorithm 2; its mean episodic return becomes its fitness. After evaluation, NEAT reproduction performs selection and topological variation to produce the next population. Across generations, the best genome among the population is tracked; after $G$ generations, the algorithm returns $g^*$.

Algorithm 2 in Appendix A evaluates one genome with online weight adaptation. For each episode, a network phenotype is created from $g$, the environment is reset, and the agent interacts for at most $T_{\max}$ steps. At step $t$, the network proposes $\hat{a}$, which is clipped to $a$ and applied to obtain $(s', r, \texttt{done})$. A reward-scaled signal $r_{\text{scaled}}$ drives plastic updates according to $\mathcal{L}$ at rate $\eta$ using only local pre and post activities together with $r_{\text{scaled}}$. The fitness is the mean return over $N$ episodes.

## 4 EXPERIMENTS

### 4.1 EXPERIMENT SETUPS

**Benchmark Environments.** We evaluate on four standard Gymnasium environments spanning diverse reward structures and action spaces. (1) `CartPole-v1` (Farama Foundation, b): control a cart on a frictionless track to balance an upright pole. The agent receives a dense reward of $+1$ per timestep until failure (the pole falls, the cart leaves the bounds, or the time limit is reached). The action space is discrete with two choices (apply force left or right), making it a low-dimensional control task. (2) `LunarLander-v2` (Farama Foundation, d): a 2D lunar module must soft land at the $(0, 0)$ pad using a main engine and two side thrusters. Rewards are densely shaped (proximity, velocity, and orientation towards landing conditions yield higher rewards; bonuses for landing legs; penalties for engine usage; terminal reward $+100$ for a safe landing and $-100$ for a crash). The action space is discrete with four choices, again a low-dimensional control task. (3) `BipedalWalker-v3` (Farama Foundation, a): a 2D Box2D biped must walk across uneven terrain (normal and hardcore variants). Rewards are dense (forward progress, $-100$ for falling, torque penalties). The action space is continuous and four-dimensional, $[-1, 1]^4$, making it a low-dimensional continuous control task. (4) `Hopper-v3` (Farama Foundation, c): a MuJoCo one-legged hopper applies torques at three joints to hop forward. The total reward is a dense combination of healthy reward, forward progress, and control cost. The action space is continuous and three-dimensional, $[-1, 1]^3$, a low-dimensional continuous control task. Environment implementations follow the Gymnasium reference; physics backends are Box2D for `LunarLander` and `BipedalWalker`, and MuJoCo for `Hopper`.

**Algorithm Protocol.** Initial policies at generation 0 are configured via NEAT with a minimal topology mapping observations directly to actions. The network topology expands during evolution using standard genetic operators of NEAT. Hidden unit activation functions are set in the configuration, while output units use `tanh`. Before activation, the summed input to each neuron is clipped to a maximum absolute value of $50.0$. For continuous control, the final `tanh` output is hard clipped to $[-1, 1]$. For discrete action spaces, the policy selects the action corresponding to the output neuron with the highest activation (`argmax`). The inner loop of online plasticity (Hebb, Oja, or BCM) is reward-modulated and runs at every step of an episode. The agent's fitness is the mean total return over $N$ evaluation episodes (where $N$ corresponds to `repeat_per_GEN` in our code). After each local weight update, the new weight value is clipped to a maximum absolute value of $10.0$. We primarily use a Lamarckian inheritance scheme in which in-episode weight updates are written back to the genome at the end of the episode (`WRITE_BACK=True`); we also include an ablation where inheritance is disabled (`WRITE_BACK=False`). The use of standard genetic weight mutation is controlled by the `WM_MODE` parameter, which can disable mutation, enable it via the configuration, or set a specific probability.

**Experimental Configuration and Evaluation.** To ensure fair comparison across population sizes $P \in \{50, 100, 200, 300\}$, we fixed a total interaction budget of $B$ environment steps per experiment. Each agent's fitness was averaged over $N$ evaluation episodes. For analysis convenience, all runs used $G = 500$ generations. For NEOL agents, the key inner loop hyperparameter (the learning rate $lr$) was selected via a grid search over $\{2.5 \times 10^{-4}, 2.5 \times 10^{-3}, 2.5 \times 10^{-2}, 2.5 \times 10^{-1}\}$. For statistical validation, each unique configuration (algorithm, hyperparameters, and population size) was evaluated over 30 independent random seeds.

### 4.2 EMPIRICAL ANALYSIS ON THE CONVERGENCE OF BEST FITNESS

To systematically evaluate the role of online neural plasticity in evolutionary processes, we conducted a comprehensive comparison against NEAT. This comparison pitted the standard NEAT method against our NEOL framework, which integrates online plasticity rules (Hebb, Oja, and

BCM). The results clearly demonstrate that the NEOL framework exhibits substantial advantages across multiple test environments.

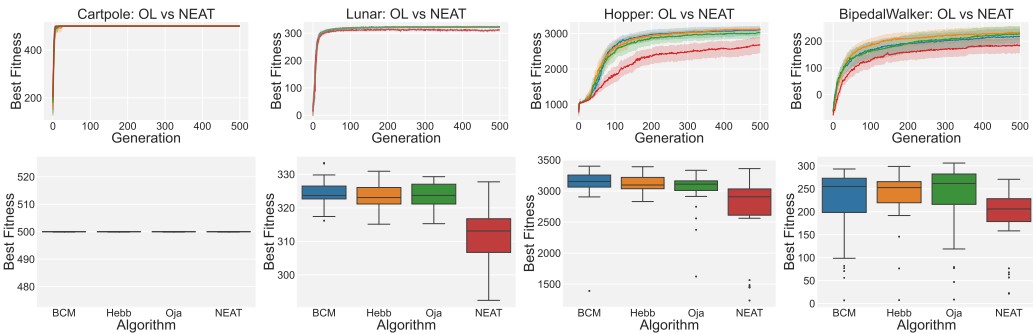

Figure 2: Performance comparison of NEOL with the standard NEAT across four environments (`CartPole-v1`, `LunarLander-v2`, `Hopper-v3`, and `BipedalWalker-v3`). **Top row**: convergence plots showing fitness over generations. **Bottom row**: boxplots of final-generation fitness distributions. BCM, Hebb, and Oja learning rules are shown in blue, orange, and green, respectively, while standard NEAT is shown in red.

In the simpler CartPole task, all methods rapidly converge to the maximum fitness, serving as a successful sanity check. However, in the more complex environments, significant performance disparities emerge. As shown in the convergence plots (Figure 2, top row), the NEOL variants consistently achieve higher final fitness scores than standard NEAT. The boxplots (Figure 2, bottom row) and standard deviations (Table 1) further reveal that while standard NEAT struggles, often resulting in a high variance and numerous low-performing outliers, the NEOL methods achieve a superior median performance. Notably, in Lunar Lander, NEOL not only outperforms NEAT but also exhibits a tighter fitness distribution, indicating higher learning reliability in terms of smaller variance.

The observed performance gains are not coincidental. We confirmed their statistical significance using a one-sided Wilcoxon rank-sum test, with the alternative hypothesis that NEOL variants achieve higher fitness than NEAT. As detailed in Table 5, we reject the null hypothesis at a significance level of $p < 0.05$ for all NEOL variants across all three complex tasks. This provides strong evidence that the integration of online learning is the source of the performance improvement.

These results lead to the conclusion that the reward-modulated online learning serves as a more effective mechanism for policy fine-tuning in NeuroEvolution than relying on structural search alone. In our framework, the outer loop of evolution discovers promising network topologies, while the inner loop of online plasticity provides an efficient, gradient-free mechanism for credit assignment and rapid weight optimisation within an agent's lifetime. This demonstrates that online plasticity is not merely an incremental add-on but can act as a powerful core component of neuroevolutionary systems, enhancing both final performance and learning reliability, even in the absence of traditional weight mutation operators.

Table 1: Best final-generation fitness (mean $\pm$ standard deviation (SD)), **Bold** marks the highest fitness per task; underline marks the runner-up.

| Task | BCM | Hebb | Oja | NEAT |
|---|---|---|---|---|
| *CartPole* | **500.00 $\pm$ 0.00** | **500.00 $\pm$ 0.00** | **500.00 $\pm$ 0.00** | **500.00 $\pm$ 0.00** |
| *Lunar Lander* | **324.34 $\pm$ 3.86** | 323.35 $\pm$ 3.67 | 323.53 $\pm$ 4.19 | 311.77 $\pm$ 8.18 |
| *Hopper* | **2983.82 $\pm$ 479.79** | 2819.89 $\pm$ 577.33 | 2900.94 $\pm$ 562.16 | 2680.22 $\pm$ 603.53 |
| *Bipedal Walker* | 217.39 $\pm$ 82.82 | **233.30 $\pm$ 62.42** | 227.10 $\pm$ 81.96 | 183.36 $\pm$ 71.82 |

## 4.3 EMPIRICAL ANALYSIS ON SAMPLE EFFICIENCY

Sample efficiency is one of the important concepts in RL algorithms (Yarats et al., 2021; D'Oro et al., 2023; Xue et al., 2024). In this paper, we also aim to evaluate the efficiency of the proposed

algorithm when interacting with any given environment in an online manner. First, we need to define a proper metric: what is sample efficiency in the context of NEOL?

Inspired and adapted from a similar learning speed metric (Peng et al., 2018) and a QD-score Area-Under-Curve (AUC) (Xue et al., 2024), we consider the following: given $M$, a total number of samples or a total number of interactions with an environment *Env* until time horizon $T$,

$$\text{SCORE} := \sum_{t=1}^{T} \frac{\text{E}_\pi(f(\pi, t))}{M} \approx \sum_{t=1}^{T} \frac{1}{M} \left( \frac{1}{n} \sum_{j=1}^{n} f(\pi_j, t) \right),$$

where $n$ is the number of independent runs we conduct and $f(\pi, t) := \sum_{\tau=0}^{t} \gamma^\tau \mathcal{R}_{Env}(s_\tau, a_\tau)$. In this paper, $n = 30$ and $M$ is the multiple of the number of generations and the population. SCORE measures how many expected cumulative rewards per sample we obtain in the training process, and it is roughly the area under the best fitness curve. It measures the optimisation efficiency of an NEOL algorithm. However, the expectation $\text{E}_\pi(f(\pi, t))$ is hard to compute in practice, and thus we use a simple Monte Carlo (Kalos & Whitlock, 2009; Owen, 2013) to approximate the value of SCORE at each time step.

Table 2: Sample Efficiency SCORE (mean $\pm$ SD) at fixed population size $\text{pop} = 300$. For plastic methods (BCM, Hebb, Oja), we use the best learning rate per task; NEAT uses its $\text{pop} = 300$ runs. Values shown with three significant figures ($a\,eb \equiv a \times 10^b$). **Bold** marks the highest value per task; underline marks the runner-up. Higher SCORE implies higher sample efficiency.

| Task | BCM | Hebb | Oja | NEAT |
|------|-----|------|-----|------|
| *CartPole* | $7.43\,\text{e}{-}1 \pm 1.09\,\text{e}{-}2$ | $\underline{7.46\,\text{e}{-}1 \pm 9.08\,\text{e}{-}3}$ | $\mathbf{7.47\,e{-}1 \pm 8.55\,e{-}3}$ | $7.44\,\text{e}{-}1 \pm 1.10\,\text{e}{-}2$ |
| *Lunar Lander* | $\underline{4.51\,\text{e}{-}1 \pm 1.22\,\text{e}{-}2}$ | $4.51\,\text{e}{-}1 \pm 6.65\,\text{e}{-}3$ | $\mathbf{4.52\,e{-}1 \pm 7.93\,e{-}3}$ | $4.37\,\text{e}{-}1 \pm 7.29\,\text{e}{-}3$ |
| *Hopper* | $\mathbf{3.77 \pm 6.26\,e{-}1}$ | $3.61 \pm 6.35\,\text{e}{-}1$ | $\underline{3.75 \pm 6.81\,\text{e}{-}1}$ | $3.22 \pm 6.73\,\text{e}{-}1$ |
| *Bipedal Walker* | $\underline{2.56\,\text{e}{-}1 \pm 1.10\,\text{e}{-}1}$ | $1.96\,\text{e}{-}1 \pm 1.39\,\text{e}{-}1$ | $\mathbf{2.60\,e{-}1 \pm 1.18\,e{-}1}$ | $1.74\,\text{e}{-}1 \pm 9.38\,\text{e}{-}2$ |

Table 2 summarises the SCORE metric at a fixed population size ($\text{pop} = 300$). Online learning via synaptic plasticity consistently improves sample efficiency over NEAT. Oja attains the best or second-best SCORE on three tasks (CartPole, Lunar Lander, Bipedal Walker), while BCM narrowly leads on Hopper with Oja a close second. The corresponding Wilcoxon rank-sum tests against NEAT (Table 6) confirm these gains: all plasticity rules are statistically significant on Lunar Lander and Hopper; on Bipedal Walker, significance holds for BCM and Oja; on CartPole, no significant differences appear owing to saturation. Overall, these results indicate that NEOL improves not only asymptotic fitness but also the sample efficiency with which good policies are discovered.

## 4.4 ABLATION STUDIES

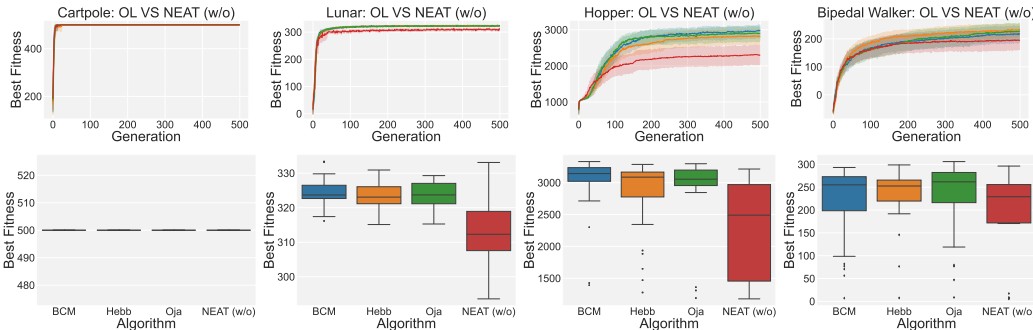

Figure 3: Ablation study comparing NEOL with NEAT (w/o), where NEAT (w/o) corresponds to disabling the weight-modulation mechanism by setting the learning rate lr = 0. **Top row:** convergence plots showing fitness over generations. **Bottom row:** boxplots of final-generation fitness distributions. BCM, Hebb, and Oja learning rules are shown in blue, orange, and green, respectively, while NEAT (w/o) is shown in red.

**Ablation on final fitness.** We ablate the online learning component by disabling weight modulation in NEAT ("NEAT (w/o)", achieved by setting the learning rate $\eta = 0$) while keeping all other settings identical. Figure 3 contrasts convergence (top row) and final-generation fitness distributions (bottom row) across CartPole, Lunar Lander, Hopper, and Bipedal Walker. In CartPole, all methods rapidly reach the performance ceiling, as expected. In Lunar Lander and Hopper, the NEOL variants (BCM, Hebb, Oja) converge faster and attain higher asymptotic fitness than NEAT (w/o). The boxplots further indicate higher medians, tighter interquartile ranges, and markedly fewer low-performing outliers for NEOL, evidencing improved stability. On the Bipedal Walker, NEOL likewise yields a higher and more reliable final performance, with reduced variance relative to NEAT (w/o). Taken together, these results isolate the contribution of online synaptic plasticity: removing it degrades both the attained fitness and the robustness of learning.

**Ablation on sample efficiency.** We also evaluate sample efficiency under the same ablation. Table 8 reports the SCORE metric, showing consistent gains for NEOL over NEAT (w/o) across the non-trivial tasks. These improvements align with the faster rise of the NEOL learning curves in Figure 3, indicating that online plasticity not only improves the final outcome but also accelerates progress towards strong policies with fewer evaluations.

**Summary.** Across tasks, the ablation confirms that the online learning mechanism is the key factor of the observed improvements: it increases asymptotic fitness, tightens performance distributions, and enhances sample efficiency, whereas removing it (NEAT (w/o)) leads to slower convergence, lower final fitness, and greater variability.

## 5 CONCLUSION AND DISCUSSION

This work addresses the core challenge of weak credit assignment in NeuroEvolution by decoupling optimisation across two timescales. We introduced NeuroEvolution Online Learning (NEOL), in which an outer generational search discovers effective network topologies via NEAT (Stanley & Miikkulainen, 2002a), while an inner, within-lifetime phase rapidly adapts synaptic weights. To propagate reward for this adaptation, we employed local, biologically plausible, reward-modulated plasticity rules: Hebb (Hebb, 1949), Oja (Oja, 1982), and BCM (Bienenstock et al., 1982) as a simple and effective mechanism for online credit assignment. To the best of our knowledge, this is a first step towards integrating online learning via synaptic plasticity into NeuroEvolution (i.e., NEAT) within a general NEOL framework. Our contribution clarifies a principled separation between structural search and local weight adaptation for credit assignment in neuroevolutionary systems.

Across benchmarks, NEOL consistently outperforms pure NEAT in final performance, reliability, and sample efficiency, with the largest gains in continuous control tasks where standard NeuroEvolution often struggles. By fixing the learning rules rather than meta-evolving them, our minimal design isolates the benefit of plasticity and contrasts with approaches that evolve the rules themselves (for example, EPANN (Soltoggio et al., 2018)) or rely on complex, global updates (for example, NEAT+RL hybrids (Peng et al., 2018)). These results indicate a strong synergy between evolutionary structural search and local online learning.

Despite the promising potential of our results, our work has some limitations. Although rigorous statistical tests were used to verify our empirical results, our proposed method lacks a theoretical guarantee, like evolutionary reinforcement learning algorithms (Lin, 2025; Qian et al., 2024; Buzdalov et al., 2013). Performance can be sensitive to task characteristics and hyperparameters, and our empirical evaluation would benefit from broader coverage to strengthen external validity.

Future work could focus on both theoretical and practical extensions of the general NEOL framework. From a theoretical perspective, it remains a challenging open problem to establish conditions under which NEOL enjoys provable convergence rates, sample efficiency bounds, or regret guarantees. On the practical side, extending NEOL with richer mutation operators, additional crossover, non-elitist selection mechanisms, and alternative synaptic plasticity rules may further enhance performance in complex settings. Scaling NEOL to larger benchmarks will enable systematic comparisons against state-of-the-art reinforcement learning and other hybrid NeuroEvolution–RL methods, yielding a clearer picture of when and how online plasticity most effectively boosts NeuroEvolution.

## ETHICS STATEMENT

We have no ethical concerns to declare.

## REPRODUCIBILITY STATEMENT

We provide an anonymous GitHub repository at `https://anonymous.4open.science/r/NeuroEvolution_Online_Learning_NEOL-41F7/Tasks/BW/ojaNEATRL.py`, containing all source code and environment specifications required to run our experiments from scratch. To preserve double blind anonymity, we do not include raw configuration files in the submission. Instead, all model/algorithm settings and training protocols (including NEOL components, optimiser choices, schedules, population size, mutation/crossover rates, selection pressure, etc.) are fully enumerated in the paper, see Section 3, Section 4, and Appendix A.2. Readers can launch from scratch with an arbitrary random seed and reproduce our tables/figures within expected stochastic variation.

**Computational Consideration:** We provide sufficient information on the computer resources used for each experiment, specifically describing the use of a general-purpose computing cluster with Ice Lake and Cascade Lake nodes. For each job, it specifies the type of compute worker (e.g., single CPU with 60+cores and 100+G memory), and the execution time (up to 12 days).

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

# A    APPENDIX

## CONTENTS

## A.1 THE USAGE OF LLM

We disclose our concrete use of large language models (LLMs) in line with the ICLR policy. LLMs are not authors; the human authors take full responsibility for all content. We mainly use LLM for writing assistance. We used LLM to polish and reorganise author-written text for academic style in academic English. Our prompt instructed the model to:

```
Correct spelling, grammar, clarity, concision, and overall
readability;
First return a polished paragraph,
then a markdown table enumerating each edit with justification;
Preserve citation strings exactly as written;
Avoid unnecessary \emph{};
Present the paragraph's intended logic before rewriting to ensure
coherence.
```

These outputs were treated as suggestions; final language and structure were decided by the authors after review. No claims, proofs, or empirical results originated from the LLM. Also, we use LLM for helping with debugging and visualisation assistance. We used LLM to (i) explain error messages, (ii) suggest small code fixes, and (iii) draft the scripts for result visualisation (e.g., plotting scripts). All suggested code was reviewed, adapted or re-implemented where non-trivial, and covered by tests. Algorithmic design, hyperparameters, and reported results were chosen by the authors.

## A.2 CONFIGURATION FOR EXPERIMENTS

### A.2.1 BEST CONFIGURATIONS (USED FOR THE FITNESS IN TABLE 1 AND TABLE 5)

Table 3: Best configurations used for the fitness results in Table 1 and Table 5. For plastic methods (BCM, Hebb, Oja), we report population size and learning rate; NEAT has no learning rate.

| Task | BCM | Hebb | Oja | NEAT |
|------|-----|------|-----|------|
| *Cartpole* | pop=50, lr=0.00025 | pop=50, lr=0.00025 | pop=50, lr=0.00025 | pop=50 |
| *Lunar Lander* | pop=300, lr=0.25 | pop=300, lr=0.00025 | pop=300, lr=0.0025 | pop=300 |
| *Hopper* | pop=300, lr=0.025 | pop=300, lr=0.00025 | pop=300, lr=0.0025 | pop=300 |
| *Bipedal Walker* | pop=300, lr=0.00025 | pop=200, lr=0.00025 | pop=300, lr=0.00025 | pop=100 |

### A.2.2 CONFIGURATIONS (USED FOR THE FITNESS IN TABLE 7 AND TABLE 8)

Table 4: Configurations used for the ablation fitness results in Table 7 and Table 8. NEAT (w/o) disables weight modulation by setting the learning rate to zero.

| Task | BCM | Hebb | Oja | NEAT (w/o) |
|------|-----|------|-----|------------|
| *Cartpole* | pop=50, lr=0.00025 | pop=50, lr=0.00025 | pop=50, lr=0.00025 | pop=50 |
| *Lunar Lander* | pop=300, lr=0.25 | pop=300, lr=0.00025 | pop=300, lr=0.0025 | pop=300 |
| *Hopper* | pop=300, lr=0.025 | pop=300, lr=0.00025 | pop=300, lr=0.0025 | pop=300 |
| *Bipedal Walker* | pop=300, lr=0.00025 | pop=300, lr=0.00025 | pop=300, lr=0.00025 | pop=300 |

## A.3 ALGORITHM PSEUDO-CODE

The online update follows one of the following local rules, where $x$ and $y$ denote pre- and post activities, $w$ is a synaptic weight, and $\theta$ is a slow activity-dependent threshold in BCM:

$$\text{WEIGHT\_UPDATE}(x, y, w, r, \beta, \eta_w) := \begin{cases} w + \eta_w \cdot \beta r \cdot y \cdot (x - y \cdot w) & \text{if reward-based Oja update} \\ w + \eta_w \cdot x \cdot y \cdot \beta r & \text{if reward-based Hebb update} \\ w + \eta_w y(y - \theta)x\beta r & \text{if reward-based BCM update} \\ w & \text{otherwise.} \end{cases}$$

---

**Algorithm 2** Online Rollout

---

Input: Genome: $g$; Learning rule: $\mathcal{L}$; Plasticity rate: $\eta$; Reward scaling factor: $\beta$; Episodes: $N$;
 Max steps: $T_{\max}$; Environment: *env*. (Defined in Algorithm 1)
Output: Average episode reward (fitness).
 1: episode_rewards = []
 2: **for** $episode \in \{1, \ldots, N\}$ **do**
 3:     net = CREATENETWORKFROMGENOME$(g)$
 4:     $s$ = env.reset()                                                             ▷ Reset state
 5:     $R_{ep} = 0$
 6:     **for** $t \in \{1, \ldots, T_{\max}\}$ **do**
 7:         $\hat{a}$ = net.forward$(s)$                          ▷ Generate the action from policy network
 8:         $a = \text{clip}(\hat{a}, -1, 1)$
 9:         $(s', r, \text{done})$ = env.step$(a)$          ▷ Get the state and reward after apply the action
 10:        $r_{scaled} = r \cdot \beta$
 11:        net.WEIGHT_UPDATE$(\mathcal{L}, \eta, r_{scaled})$          ▷ Update weight with specific learning rule
 12:        $R_{ep} = R_{ep} + r$
 13:        $s = s'$
 14:        **if** done **then break**
 15:     episode_rewards.append$(R_{ep})$
 16: **return** Average cumulative episode rewards

---

**Algorithm 3** Sepciate (NEAT) (Stanley & Miikkulainen, 2002a)

---

Input: Population with raw fitness: $\mathcal{P}$; NEAT config $\Theta_{\text{NEAT}}$ (compatibility coeffs $c_1, c_2, c_3$, threshold $\delta_t$, survival threshold $\rho$, elitism $E$, stagnation limit $G_{\text{stag}}$, min_species_size, etc.).
Output: Species set $\mathcal{S}$; population $\mathcal{P}$ with *adjusted* fitness.
 1: $\mathcal{S} = \emptyset$                                    ▷ if previous species exist, reuse their representatives
 2: **for** genome $g \in \mathcal{P}$ **do**
 3:     Compute compatibility distance to each representative $r_s$:

$$\delta(g, r_s) = c_1 \frac{E + D}{N} + c_3 \cdot \overline{|w_g - w_{r_s}|}$$

 4:     Assign $g$ to $\arg\min_s \delta(g, r_s)$ if $\min_s \delta \le \delta_t$; else create a new species with $g$.
 5: **for** species $s \in \mathcal{S}$ **do**
 6:     Sort members by raw fitness (desc), record champion.
 7:     Update $s$'s best-so-far and stagnant counter; mark for removal if $> G_{\text{stag}}$ (optionally keep global best).
 8: Remove stagnant species; ensure each remaining species has $\ge 1$ member.
 9: (**Explicit sharing**) For each species $s$ and $g \in s$:

$$f^{\text{adj}}(g) = \frac{f(g)}{|s|}, \qquad F^{\text{adj}}(s) = \frac{1}{|s|} \sum_{g \in s} f^{\text{adj}}(g).$$

 10: **return** $(\mathcal{S}, \mathcal{P})$.

---

Algorithm 3 performs speciation and fitness adjustment for NEAT. Genomes are assigned to species by compatibility distance with coefficients $c_1, c_2, c_3$ and threshold $\delta_t$. Using excess and disjoint gene counts $E$ and $D$, the number of matched genes $N$, and the mean absolute weight difference,

---

**Algorithm 4** Reproduce (NEAT) (Stanley & Miikkulainen, 2002a)

---

**Require:** Species set $\mathcal{S}$ and adjusted fitness $F^{\text{adj}}(s)$ computed by Alg. 3.

Input: Population $\mathcal{P}$ with adjusted fitness; NEAT config $\Theta_0$ (elitism $E$, survival threshold $\rho$, crossover prob. $p_c$, add-connection $m_\ell$, add-node $m_n$, weight-mutation mode WM_MODE); target size $P$.

Output: Next-generation population $\mathcal{P}_{\text{new}}$.

1:   $A = \sum_{s \in \mathcal{S}} F^{\text{adj}}(s)$
2: **for** each $s \in \mathcal{S}$ **do**
3:       spawn$(s) = \max(\text{min\_species\_size}, \text{round}(P \cdot F^{\text{adj}}(s)/A))$
4: Renormalise spawn$(\cdot)$ so that $\sum_s \text{spawn}(s) = P$
5: $\mathcal{P}_{\text{new}} = \emptyset$
6: **for** each species $s \in \mathcal{S}$ **do**
7:       Sort members of $s$ by raw fitness (descending)
8:       Copy top $E$ elites of $s$ to $\mathcal{P}_{\text{new}}$
9:       spawn$(s) = \text{spawn}(s) - E$
10:     **if** spawn$(s) > 0$ **then**
11:         $K = \lceil \rho \cdot |s| \rceil$
12:         $U = \text{top-}K$ members of $s$         ▷ parent pool
13:         **while** spawn$(s) > 0$ **do**
14:            spawn$(s) = \text{spawn}(s) - 1$
15:            Sample $p_1 \sim U$
16:            With probability $p_c$, sample $p_2 \sim U$; otherwise set $p_2 = p_1$
17:            offspring $=$ CROSSOVERALIGNED$(p_1, p_2)$    ▷ align by innovation numbers
18:            **if** rand $< m_\ell$ **then**
19:                ADDCONNECTION(offspring)
20:            **if** rand $< m_n$ **then**
21:                ADDNODE(offspring)
22:            **if** WM_MODE $=$ off **then**
23:                NOWEIGHTMUTATION(offspring)
24:            **else if** WM_MODE $=$ config **then**
25:                MUTATEWEIGHTSBYCONFIG(offspring, $\Theta_0$)
26:            **else**
27:                MUTATEWEIGHTSWITHPROB(offspring, $p$)
28:            Append offspring to $\mathcal{P}_{\text{new}}$
29: **return** $\mathcal{P}_{\text{new}}$

---

the distance to a species representative $r_s$ is

$$\delta(g, r_s) = c_1 \frac{E + D}{N} + c_3 \overline{|w_g - w_{r_s}|}.$$

Within each species, members are ranked by raw fitness, champions are tracked, and species that stagnate beyond $G_{\text{stag}}$ may be removed except for a possible global best safeguard. Explicit fitness sharing is applied:

$$f^{\text{adj}}(g) = \frac{f(g)}{|s|}, \qquad F^{\text{adj}}(s) = \frac{1}{|s|} \sum_{g \in s} f^{\text{adj}}(g).$$

Algorithm 4 generates the next population under speciated reproduction. Let $A = \sum_s F^{\text{adj}}(s)$ be the sum of adjusted fitness across species. Each species receives an offspring budget

$$\text{spawn}(s) = \max\left(\text{min\_species}, \text{round}\left(P \frac{F^{\text{adj}}(s)}{A}\right)\right),$$

renormalised so that the counts sum to $P$. Elites $E$ are copied unchanged. The remaining offspring are bred from the top $\rho$ fraction within each species. Parents are selected, crossover is applied with probability $p_c$ using historical innovation numbers for alignment, and structural mutations are applied with probabilities $m_\ell$ (add connection) and $m_n$ (add node). Weight mutation is controlled

---

by `WM_MODE`, which can disable weight mutation, use configuration defaults, or apply a specified probability $p$. The resulting offspring across all species form the next population consumed by Algorithm 1.

### A.4 STATISTICAL TESTS FOR COMPARISON ON BEST FITNESS AND SAMPLE EFFICIENCY SCORE

Table 5: Wilcoxon rank-sum test: $p$-values for Bipedal Walker, Hopper, and Lunar Lander, comparing each OL method (BCM, Hebb, Oja) against NEAT. For each task and method, we use the best configuration selected by means of final-generation best across 30 seeds, and test the final best values. The null hypothesis is that the two configurations yield samples from the same distribution; the alternative hypothesis is that the OL method tends to achieve the larger best-fitness than NEAT. **Bold** entries indicate $p < 0.05$ (rejecting the null at the 5% level). Cartpole is omitted because all methods achieve 500.0 exactly in the end. Values shown with three significant figures ($a\,eb \equiv a \times 10^b$).

| Task | BCM vs NEAT | Hebb vs NEAT | Oja vs NEAT |
|---|---|---|---|
| *Lunar Lander* | **1.202 $e-8$** | **3.352 $e-8$** | **4.686 $e-8$** |
| *Hopper* | **1.680 $e-3$** | 5.746 e-2 | **1.501 $e-2$** |
| *Bipedal Walker* | **4.033 $e-3$** | **3.006 $e-4$** | **1.004 $e-3$** |

Table 6: Wilcoxon rank-sum test: $p$-values for CartPole, Bipedal Walker, Hopper, and Lunar Lander, comparing each OL method (BCM, Hebb, Oja) against NEAT. For each task and method, we use the best configuration selected by means of final-generation best across 30 seeds, and test the final best values. The null hypothesis is that the two configurations yield samples from the same distribution; the alternative hypothesis is that the OL method tends to achieve *larger* best-fitness than NEAT. **Bold** entries indicate $p < 0.05$ (rejecting the null at the 5% level). Values shown with three significant figures ($a\,eb \equiv a \times 10^b$).

| Task | BCM vs NEAT | Hebb vs NEAT | Oja vs NEAT |
|---|---|---|---|
| *CartPole* | 5.19 e-1 | 6.62 e-1 | 5.93 e-1 |
| *Lunar Lander* | **1.20 $e-8$** | **2.03 $e-9$** | **2.44 $e-9$** |
| *Hopper* | **1.89 $e-4$** | **8.68 $e-3$** | **5.61 $e-5$** |
| *Bipedal Walker* | **9.52 $e-4$** | 3.95 e-1 | **1.11 $e-3$** |

## A.5 Additional Tables for Ablation Studies

In this ablation study, we keep NEOL unchanged and ablate only the pure NEAT baseline by disabling genetic weight mutation. The ablated counterpart is denoted NEAT (w/o). This setting is equivalent to setting the online rate of online learning to zero (fixing the weight mutation of NEAT means there is only evolutionary topology search remaining), and isolates whether online plasticity inside NEOL can substitute for, or complement, evolutionary weight mutation.

From Table 7, we compare standard NEOL (BCM, Hebb, Oja) with the ablated pure NEAT baseline (NEAT (w/o)). On Cartpole, all methods reach the optimum and are indistinguishable. On Lunar Lander, every NEOL rule yields a higher mean than NEAT (w/o), with BCM at 324.34, Oja at 323.53, and Hebb at 323.35, versus 312.50 for NEAT (w/o); the standard deviation for NEOL is small, indicating reliable convergence. On Hopper, the gap is significant: BCM, Oja, and Hebb achieve 2983.82, 2900.94, and 2819.89, respectively, compared with 2296.57 for NEAT (w/o), with difference shown and well-separated means. On Bipedal Walker, NEOL still leads in the mean (Hebb 233.30, Oja 227.10, BCM 217.39) over NEAT (w/o) (194.91), but standard deviations are wide for all configurations, suggesting that although Bipedal Walker is a harder task for all the algorithms, every NEOL is still robust. With weight mutation disabled in the counterpart, reward-modulated online plasticity is sufficient to recover and surpass final optimisation on Lunar Lander and Hopper tasks.

Table 7: Ablation study comparing NEOL with NEAT (w/o) on best final-generation fitness (mean $\pm$ SD) for each task and method using each method's best hyperparameters. NEAT (w/o) corresponds to disabling the weight-modulation mechanism by setting the learning rate lr $= 0$. **Bold** marks the highest fitness per task; underline marks the runner-up.

| Task | BCM | Hebb | Oja | NEAT (w/o) |
|---|---|---|---|---|
| *Cartpole* | **500.00 $\pm$ 0.00** | **500.00 $\pm$ 0.00** | **500.00 $\pm$ 0.00** | **500.00 $\pm$ 0.00** |
| *Lunar Lander* | **324.34 $\pm$ 3.86** | 323.35 $\pm$ 3.67 | 323.53 $\pm$ 4.19 | 312.50 $\pm$ 8.97 |
| *Hopper* | **2983.82 $\pm$ 479.79** | 2819.89 $\pm$ 577.33 | 2900.94 $\pm$ 562.16 | 2296.57 $\pm$ 740.56 |
| *Bipedal Walker* | 217.39 $\pm$ 82.82 | **233.30 $\pm$ 62.42** | 227.10 $\pm$ 81.96 | 194.91 $\pm$ 92.83 |

From Table 8, we compare standard NEOL (BCM, Hebb, Oja) with the ablated NEAT baseline without weight mutation (NEAT (w/o)) on SCORE. On CartPole, all methods are effectively indistinguishable, with Oja slightly highest ($7.47\times10^{-1}$) and NEAT (w/o) a close second ($7.46\times10^{-1}$). On Lunar Lander, every NEOL rule improves over NEAT (w/o) ($4.29\times10^{-1}$), with Oja best ($4.52\times10^{-1}$) and BCM runner-up ($4.51\times10^{-1}$); the values of standard deviation are small across NEOL (except BCM), indicating stable gains. On Hopper, the advantage is obvious: BCM achieves the top SCORE ($3.77 \pm 6.26\times10^{-1}$) with Oja close behind ($3.75 \pm 6.81\times10^{-1}$), both well above NEAT (w/o) ($3.05\pm9.33\times10^{-1}$); standard deviations are moderate but the means remain separated. On Bipedal Walker, Oja leads ($2.60\times10^{-1}$) with BCM runner-up ($2.56\times10^{-1}$) and NEAT (w/o) trailing ($2.43\times10^{-1}$); advantage is substantial for all methods (except Hebb), suggesting promising potential for online learning methods. Overall, even when evolutionary weight mutation is disabled in the counterpart, reward-modulated online plasticity in NEOL improves sample efficiency on the non-trivial tasks, while CartPole remains saturated.

Table 8: Ablation study comparing NEOL with NEAT (w/o) on sample efficiency SCORE (mean $\pm$ SD) for each task and method using each method's best hyperparameters. NEAT (w/o) corresponds to disabling the weight-modulation mechanism by setting the learning rate lr $= 0$. **Bold** marks the highest fitness per task; underline marks the runner-up.

| Task | BCM | Hebb | Oja | NEAT (w/o) |
|---|---|---|---|---|
| *Cartpole* | $7.43\,e{-}1 \pm 1.09\,e{-}2$ | $7.46\,e{-}1 \pm 9.09\,e{-}3$ | **$7.47\,e{-}1 \pm 8.55\,e{-}3$** | $7.46\,e{-}1 \pm 9.59\,e{-}3$ |
| *Lunar Lander* | $4.51\,e{-}1 \pm 1.22\,e{-}2$ | $4.51\,e{-}1 \pm 6.65\,e{-}3$ | **$4.52\,e{-}1 \pm 7.93\,e{-}3$** | $4.29\,e{-}1 \pm 1.36\,e{-}2$ |
| *Hopper* | **$3.77 \pm 6.26\,e{-}1$** | $3.61 \pm 6.35\,e{-}1$ | $3.75 \pm 6.81\,e{-}1$ | $3.05 \pm 9.33\,e{-}1$ |
| *Bipedal Walker* | $2.56\,e{-}1 \pm 1.10\,e{-}1$ | $1.96\,e{-}1 \pm 1.39\,e{-}1$ | **$2.60\,e{-}1 \pm 1.18\,e{-}1$** | $2.43\,e{-}1 \pm 1.18\,e{-}1$ |

## A.6 MORE EXPERIMENT RESULTS ON THE FINAL BEST FITNESS

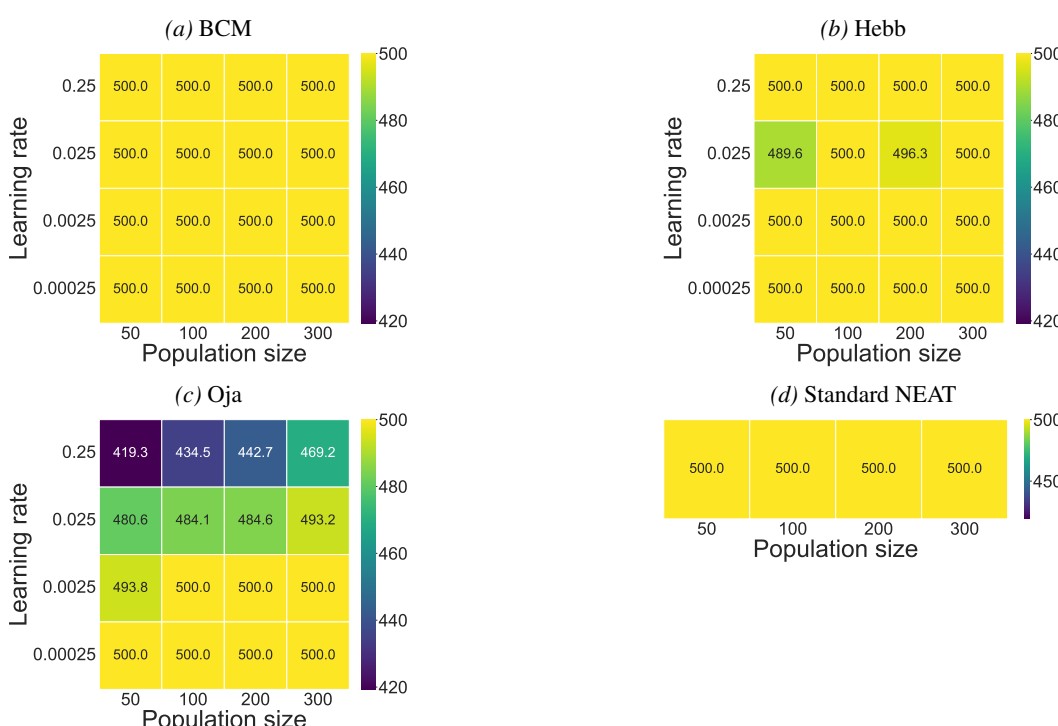

Figure 4: Heatmap comparison of NEOL algorithms (BCM, Hebb, Oja) against standard NEAT in `CartPole-v1`. Values represent the empirical mean of the final best fitness over 30 seeds.

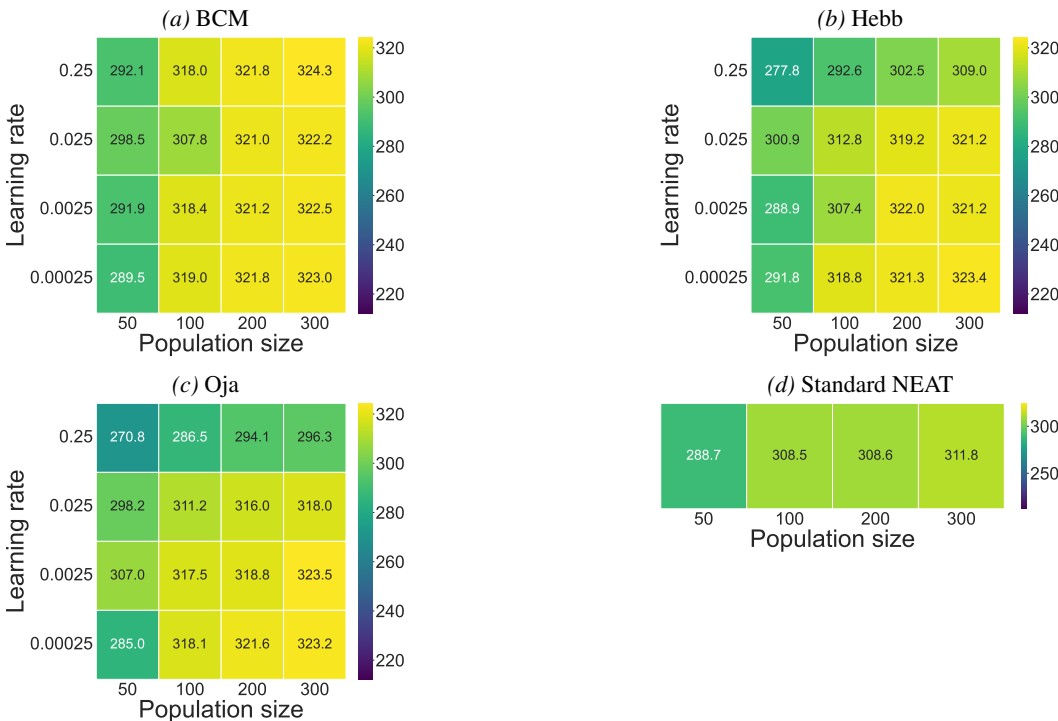

Figure 5: Heatmap comparison of NEOL algorithms (BCM, Hebb, Oja) against standard NEAT in `LunarLander-v2`. Values represent the empirical mean of the final best fitness over 30 seeds.

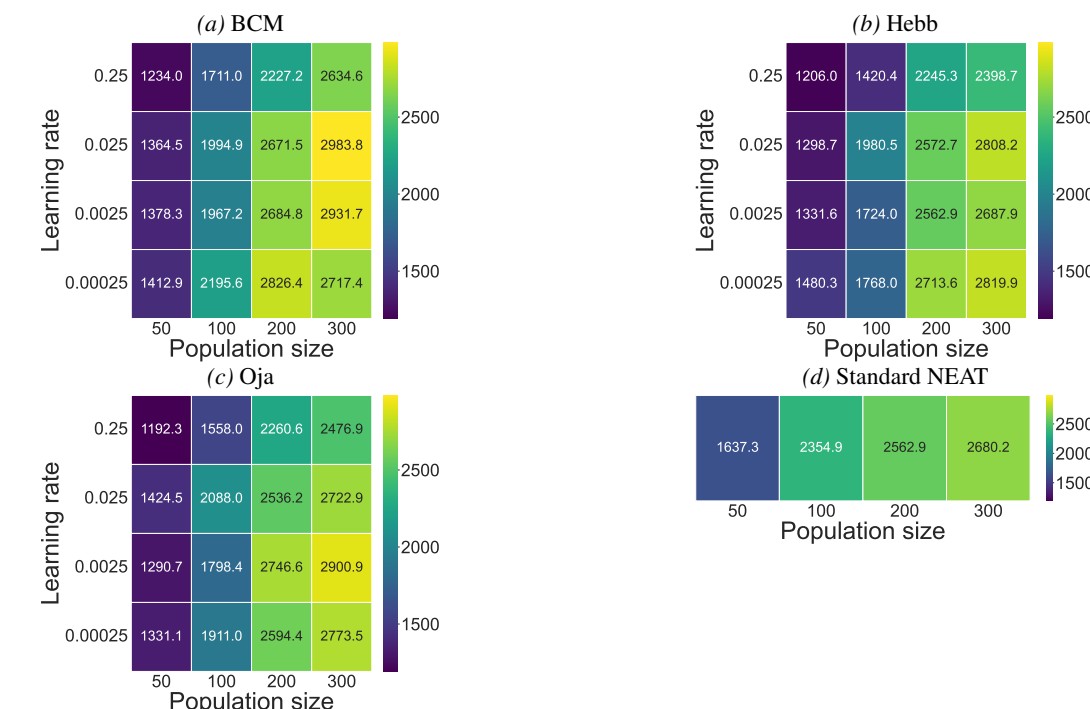

Figure 6: Heatmap comparison of NEOL algorithms (BCM, Hebb, Oja) against standard NEAT in `Hopper-v3`. Values represent the empirical mean of the final best fitness over 30 seeds.

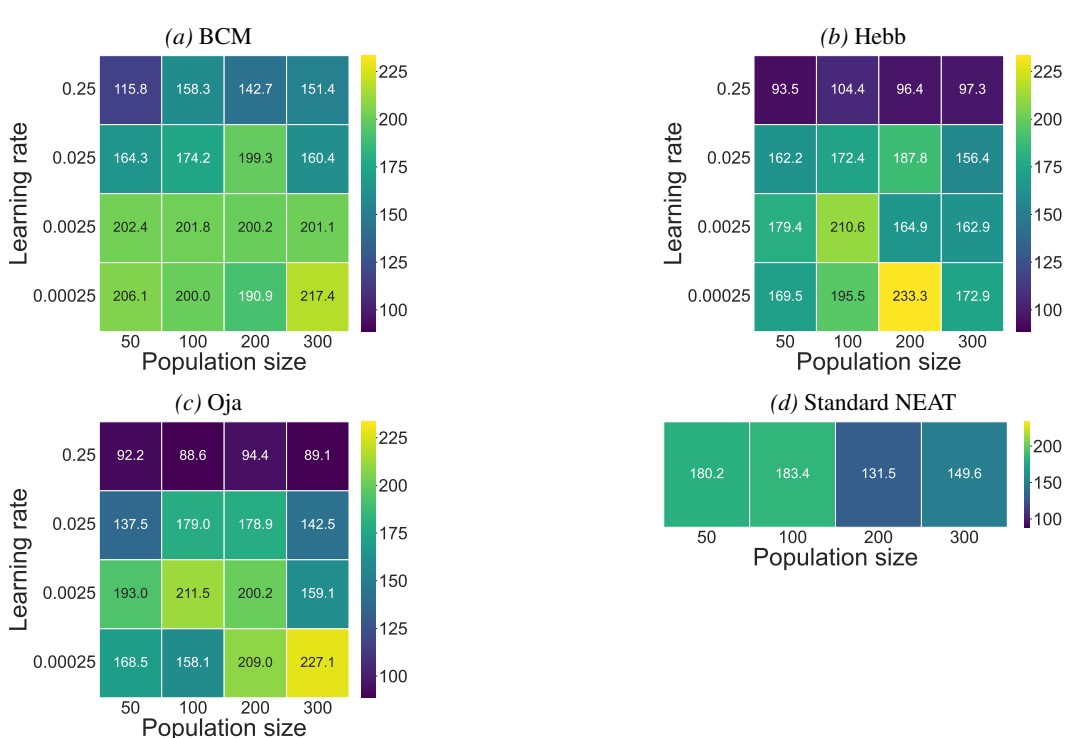

Figure 7: Heatmap comparison of NEOL algorithms (BCM, Hebb, Oja) against standard NEAT in `BipedalWalker-v3`. Values represent the empirical mean of the final best fitness over 30 seeds.

## A.7    MORE EXPERIMENT RESULTS ON THE SAMPLE EFFICIENCY SCORE

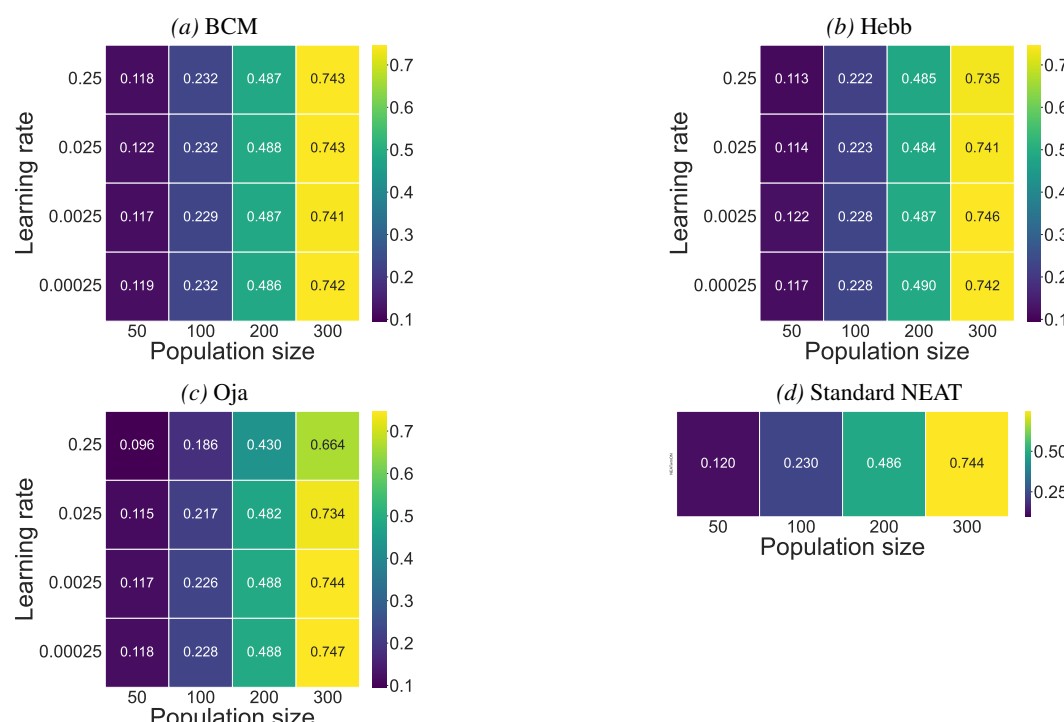

Figure 8: Heatmap comparison of NEOL algorithms (BCM, Hebb, Oja) against standard NEAT in `CartPole-v1`. Values represent the Sample Efficiency SCORE over 30 seeds.

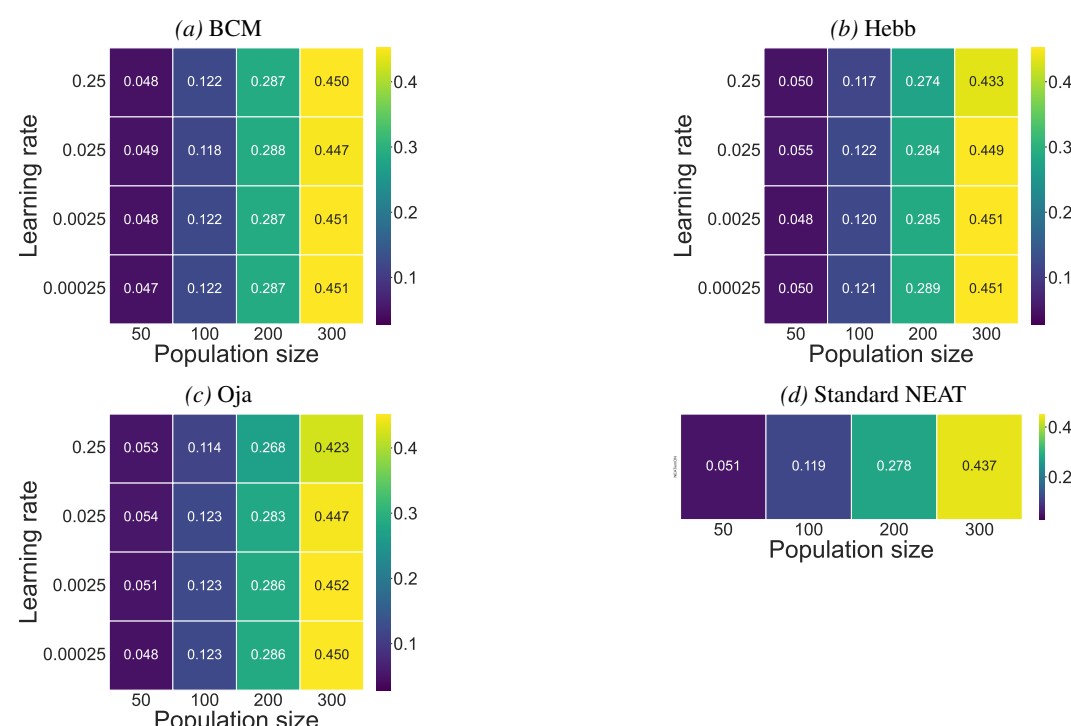

Figure 9: Heatmap comparison of NEOL algorithms (BCM, Hebb, Oja) against standard NEAT in `LunarLander-v2`. Values represent the Sample Efficiency SCORE over 30 seeds.

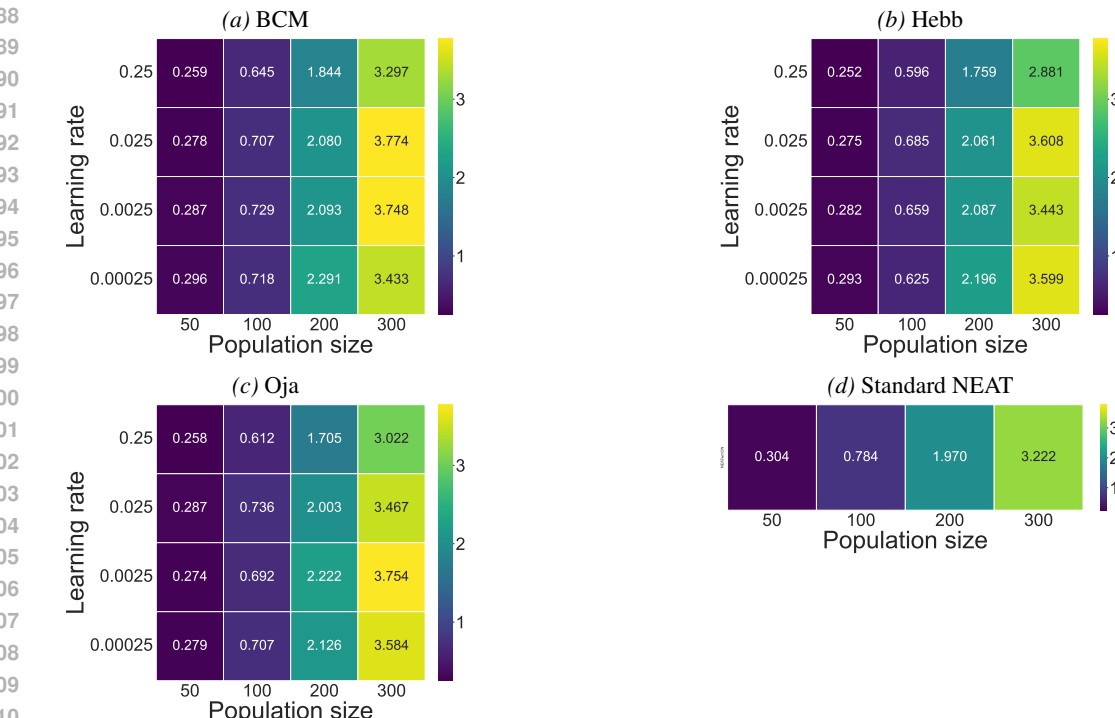

Figure 10: Heatmap comparison of NEOL algorithms (BCM, Hebb, Oja) against standard NEAT in `Hopper-v3`. Values represent the Sample Efficiency SCORE over 30 seeds.

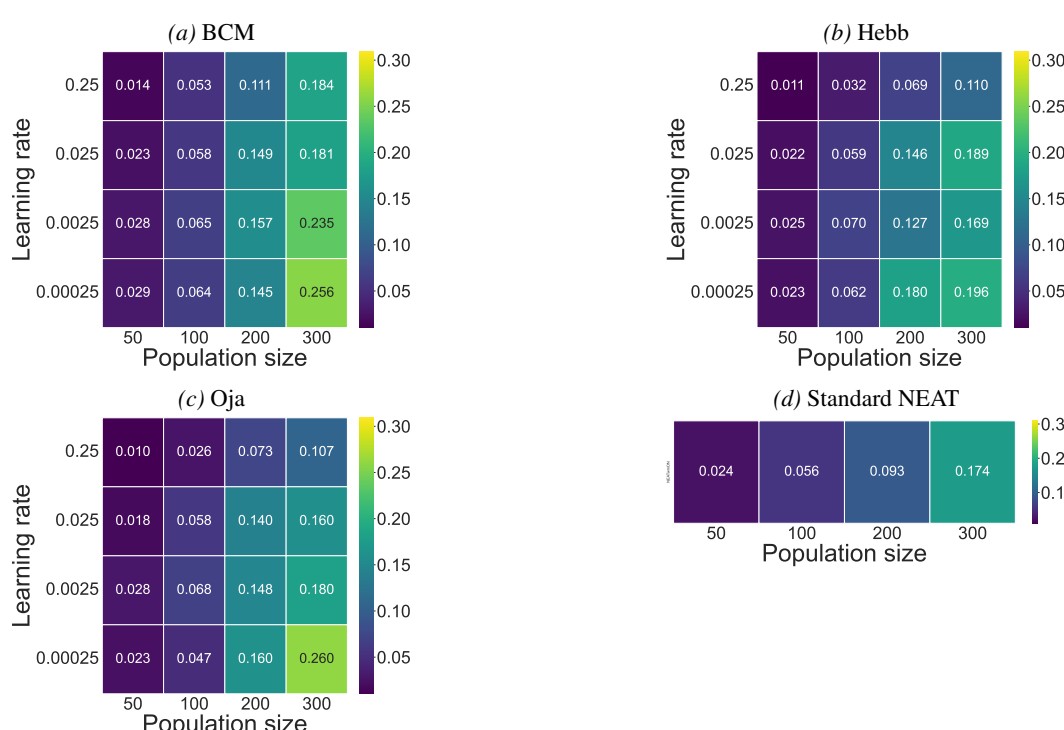

Figure 11: Heatmap comparison of NEOL algorithms (BCM, Hebb, Oja) against standard NEAT in `BipedalWalker-v3`. Values represent the Sample Efficiency SCORE over 30 seeds.

A.8 Summary of More Experiment Results

**Parameter Setting.** The heatmaps in Figure 4,Figure 5,Figure 6,Figure 7, Figure 8, Figure 9, Figure 10, Figure 11 report a parameter sweep over population size $\{50, 100, 200, 300\}$ and learning rates $\{2.5 \times 10^{-4}, 2.5 \times 10^{-3}, 2.5 \times 10^{-2}, 2.5 \times 10^{-1}\}$ for the three NEOL variants (BCM, Hebb, Oja) alongside the standard NEAT baseline. Each pixel shows the empirical mean of the final best fitness over 30 seeds; the vertical axis is the plasticity learning rate, and the horizontal axis is the population size. Because NEAT has no learning rate, it appears as a single row per environment.

A.8.1 Best Fitness

On CartPole-v1 (Figure 4). All methods reach the task ceiling and remain flat across most settings in best fitness. BCM and Hebb saturate at 500 for every population size and plasticity rate on our heatmap, indicating that structural search alone or in combination with offline updates is sufficient on this easy, discrete–action benchmark. Oja exhibits a mild instability only at the largest plasticity rates and smallest populations (top row, leftmost columns), where the mean final best fitness drops below the ceiling, but recovers as the rate is reduced or the population increases. The NEAT baseline sits at the ceiling for all population sizes.

On LunarLander-v2 (Figure 5). The three NEOL variants consistently dominate the NEAT row and exhibit a smooth improvement with population size. BCM is the most robust to the plasticity learning rate: means above 320 are obtained for populations 200–300 across a wide range of rates, and the best cell is attained at population 300. Hebb is more sensitive to the learning rate: very large learning rates combined with small populations depress performance, while learning rates in $[2.5 \times 10^{-4}, 2.5 \times 10^{-3}]$ recover and surpass 320 as the population grows. Oja shows a similar pattern, with its best region again in the bottom half of the grid and larger populations. The NEAT row improves slightly with population, but remains roughly 10–15 points below the strongest NEOL settings.

On Hopper-v3 (Figure 6). For BCM and Oja the surface rises sharply with population and peaks at intermediate plasticity learning rates ($2.5 \times 10^{-3}$ or $2.5 \times 10^{-2}$), reaching mean final best fitness near or above 2.9k at population 300. Very small learning rates underfit and very large rates overfit or destabilise, producing a characteristic ridge across the middle rows. Hebb benefits from the same scaling trends but remains below BCM and Oja over most of the grid, particularly at small populations or extreme rates. Standard NEAT also scales with population but plateaus several hundred points below the best NEOL cells, indicating that online weight adaptation contributes materially beyond structural search in this domain. Continuous control displays a pronounced interaction between population size and plasticity rate.

On BipedalWalker-v3 (Figure 7). NEOL improves upon NEAT across broad regions. Hebb attains the highest cell in the grid at population 200 with the smallest plasticity learning rate, and degrades rapidly as the learning rate increases, especially at small populations. Oja and BCM display more gradual trends: performance climbs with population size and is best in the lowest–rate row, with Oja's peak at population 300 and BCM's at population 300 as well. The NEAT row is comparatively flat and non–monotonic in population, with means concentrated around 130–180 and no configuration matching the top NEOL cells. These results suggest that modest, reward–gated plasticity combined with sufficient population–level exploration is beneficial, whereas aggressive learning rates are detrimental in this environment.

A.8.2 Sample Efficiency

On CartPole-v1 (Figure 8). All methods are highly sample–efficient once population size is moderate, with scores climbing toward the upper bound of the metric. BCM and Hebb form broad plateaus that peak at population 300 (around $0.74$), matching or slightly edging the NEAT row. Oja shows a mild degradation only at the largest plasticity rates and smallest populations (top–left cells), but recovers as the rate is reduced or the population increases. The NEAT baseline improves steadily with population (from $\approx 0.12$ at 50 to $\approx 0.74$ at 300).

On LunarLander-v2 (Figure 9). The three NEOL variants consistently dominate the NEAT row at matched populations and improve smoothly with population size. BCM shares a similar pattern to the other two methods: very large rates coupled with small populations reduce efficiency, while rates

in $[2.5 \times 10^{-4}, 2.5 \times 10^{-3}]$ recover as the population grows. NEAT improves with population but remains a few points below the strongest NEOL settings (peaking around $\approx 0.437$).

On Hopper-v3 (Figure 10). BCM and Oja exhibit steep gains with population and peak at intermediate plasticity learning rates ($2.5 \times 10^{-3}$–$2.5 \times 10^{-2}$), reaching the highest sample–efficiency scores in the grid (near $\sim 3.7$ at population 300). Very small learning rates underfit and very large rates destabilise, producing a ridge of best performance across the middle rows. Hebb follows the same scaling trend but remains below BCM and Oja over most of the grid. Standard NEAT also benefits from larger populations yet plateaus well below the best NEOL cells (around $\sim 3.2$), indicating a clear advantage from online weight adaptation beyond standard offline search.

On BipedalWalker-v3 (Figure 11). NEOL improves upon NEAT across broad regions, with absolute scores lower than Hopper but similar relative trends. Oja and BCM increase steadily with population and perform best in the lowest–rate row, peaking at population 300 (Oja $\approx 0.26$, BCM $\approx 0.26$). Hebb reaches its best cells only with low rates and larger populations and degrades rapidly as the learning rate increases, especially at small populations. The NEAT row is comparatively flat and never reaches the top NEOL cells (max $\approx 0.17$).

Across environments, larger populations consistently improve both sample efficiency and final best fitness for NEOL, reflecting stronger structural exploration and better coverage of favourable topologies that can then be fine–tuned online. Modest, reward–gated plasticity further strengthens these gains, whereas aggressive learning rates are either unnecessary on saturated, discrete tasks or harmful on fragile continuous–control tasks. Among plasticity rules, BCM and Oja yield the smoothest and most robust behaviour on the continuous–control benchmarks, while Hebb is markedly more rate–sensitive. The single–row NEAT baseline trails the best NEOL settings wherever the task is not trivially solved, supporting the claim that reward–modulated online weight adaptation complements evolutionary topology search.

