# OpenReview forum: "NEOL: REWARD-GATED ONLINE PLASTICITY FOR SCALABLE NEUROEVOLUTION"
_ICLR.cc/2026/Conference — ICLR 2026 Conference Withdrawn Submission_

### Official Review · Reviewer_bgzr · 2025-10-27

**Soundness:** 2
**Presentation:** 1
**Contribution:** 1
**Rating:** 2
**Confidence:** 4

**Summary:**

This paper proposes a method for simultaneously evolving neural network topology and synaptic weights using an outer-loop/inner-loop framework, and evaluates it on four benchmark tasks.

**Strengths:**

- The idea of co-evolving topology and synaptic weights in a nested-loop framework is conceptually interesting and, to the best of my knowledge, relatively underexplored in the current literature.

**Weaknesses:**

1. **Misalignment of the Outer-Loop/Inner-Loop Framework:**
While the separation of topology evolution (outer loop) and weight adaptation (inner loop) is structurally reasonable, the experimental design raises concerns about the authors’ understanding of the fundamental purpose of such a nested-loop architecture. Typically, the outer loop is expected to optimize meta-parameters (e.g., network topology, learning rules, or plasticity coefficients) across a *distribution* of tasks, so that the inner loop can generalize to *new, related tasks* using the optimized meta-knowledge. However, this paper applies the framework to a *single, stationary task*, which undermines the very rationale for using an outer-loop/inner-loop structure. This design choice appears conceptually flawed and limits the significance of the results.

2. **Over-Simplification of Synaptic Plasticity:**
The paper adopts a *fixed, homogeneous* plasticity rule (i.e., a single, hand-tuned learning rate η) across all synapses. This is a significant oversimplification. A core objective in many prior works is to *evolve* plasticity rules or their hyperparameters (e.g., learning rates, modulatory signals) in the outer loop, enabling task-specific adaptation. The use of a uniform, human-specified plasticity coefficient not only reduces biological plausibility but also limits the adaptability and expressiveness of the model. The contribution thus risks appearing as an ad-hoc combination of topology evolution and plasticity (i.e., “A+B”) without a principled integration.

3. **Limited and Weak Baselines:**
   The empirical evaluation lacks breadth and depth. Several relevant and recent works are omitted, particularly those that integrate plasticity with meta-learning, recurrent memory, or neuromodulation. For instance:
   - [1]: Growing with Experience: Growing Neural Networks in Deep Reinforcement Learning
   - [2]: Neuroplastic Expansion in Deep Reinforcement Learning
   - [3] Soltoggio, Andrea, et al. "Evolutionary advantages of neuromodulated plasticity in dynamic, reward-based scenarios." Proceedings of the 11th international conference on artificial life (Alife XI). MIT Press, 2008.
   - [4] Mishra, Nikhil, et al. "A Simple Neural Attentive Meta-Learner." International Conference on Learning Representations. 2018.
   - [5] Joachim Winther Pedersen and Sebastian Risi. Evolving and merging hebbian learning rules: increasing generalization by decreasing the number of rules. In Proceedings of the Genetic and Evolutionary Computation Conference, pp. 892–900, 2021.
   - [6] Wang, Fan, et al. "Evolving Decomposed Plasticity Rules for Information-Bottlenecked Meta-Learning." Transactions on Machine Learning Research.

4. The paper requires substantial revision for clarity and precision. For example:
   - The definition of “M” as “a total number of samples or a total number of interactions with an environment Env until time horizon T” is ambiguous. Please clarify whether M refers to episodes, steps, or transitions, and whether it is fixed or task-dependent.
   - The term “credit assignment” is used repeatedly (e.g., “reward-gated and behaviorally relevant credit assignment”), but its technical meaning is unclear in context. Is this referring to temporal credit assignment in RL, or to a biologically inspired learning signal? If the latter, please explicitly connect it to the plasticity rule and justify its relevance.

**Questions:**

See the weaknesses

---

### Official Review · Reviewer_q1Gg · 2025-10-31

**Soundness:** 2
**Presentation:** 1
**Contribution:** 2
**Rating:** 4
**Confidence:** 4

**Summary:**

The authors propose NeuroEvolutionary Online Learning (NEOL), a hybrid framework that explicitly decouples structural evolution from weight adaptation. In NEOL, an outer loop employs standard NEAT for topological search, while a novel inner loop performs online, within-episode weight adaptation using reward-modulated local synaptic plasticity rules (specifically Hebbian, Oja's, and BCM). The method is evaluated on four classic control benchmarks (CartPole, LunarLander, BipedalWalker, Hopper), comparing the NEOL variants against a standard NEAT baseline. The results demonstrate that NEOL achieves statistically significant improvements in final return, sample efficiency (as measured by a custom SCORE metric), and solution robustness (lower variance), with gains being most pronounced in the continuous control tasks.

**Strengths:**

- The experimental comparison to NEAT is thorough. The use of 30 random seeds, multiple benchmarks spanning discrete and continuous action spaces, and appropriate statistical testing (Wilcoxon rank-sum) provides strong evidence for the central claim: that NEOL is a superior alternative to standard NEAT.

**Weaknesses:**

- The paper's motivation rests on NEAT's poor scaling, a problem that other methods (e.g., HyperNEAT, NEAT-PGS, modern Evolution Strategies) also purport to solve. More importantly, to position NEOL as a practical and relevant algorithm, it must be compared against standard gradient-based RL algorithms (e.g., PPO, SAC) or at least modern gradient-free methods (e.g., Salimans et al., 2017) on the same continuous control tasks. Without this context, it is impossible to know if NEOL is a competitive learning algorithm in 2026 or merely a better version of NEAT.

- While the specific implementation within NEAT may be novel, the high-level concept of a two-timescale system (outer-loop evolution, inner-loop plasticity/learning) is a foundational concept in the field (e.g., the Baldwin effect, and more directly, the extensive work on evolving plastic ANNs cited by the authors, such as Soltoggio et al., 2008, and Najarro & Risi, 2020). The plasticity rules (Hebb, Oja, BCM) and their reward-modulation (Frémaux & Gerstner, 2016) are also pre-existing. The paper's contribution is more an effective engineering integration and rigorous comparison rather than a fundamental mechanistic breakthrough

**Questions:**

1. The paper states it primarily uses a Lamarckian scheme (WRITE_BACK=True, Line 309), where adapted weights are inherited. An ablation (WRITE_BACK=False) is mentioned but no data is presented. How critical is this Lamarckian property? Does a purely Darwinian approach (where plasticity is only for evaluation fitness, but weights are not written back to the genome) also achieve significant gains over NEAT? This is a key mechanistic question.

2. How does NEOL's final performance and, critically, its sample efficiency (in wall-clock time or total environment steps) compare to well-tuned implementations of PPO or SAC on the Hopper-v3 and BipedalWalker-v3 tasks? This context is essential for positioning the work.

3. Could you clarify the "fixed total interaction budget B" (Line 317)? The SCORE metric (Eq 4) is an AUC, which is dependent on the total time horizon $T$ (or total samples $M$). How was $B$ used to normalize runs with different population sizes (e.g., $P=50$ vs. $P=300$)? Does a larger population run for fewer generations to maintain the same $B$? This is unclear.

---

### Official Review · Reviewer_XEuE · 2025-11-01

**Soundness:** 3
**Presentation:** 3
**Contribution:** 2
**Rating:** 2
**Confidence:** 4

**Summary:**

This paper introduces NEOL, a hybrid framework that integrates reward-modulated local plasticity into the NEAT algorithm. The central idea is to decouple topology evolution (handled by NEAT) from weight adaptation (handled by fixed online learning rules such as Hebbian, Oja, or BCM).

During each episode, synaptic weights adapt online according to a biologically inspired rule gated by a reward signal. After the episode, cumulative reward is used as the fitness signal for evolution. The authors show that NEOL improves convergence speed, fitness stability, and final performance over standard NEAT across several classic control benchmarks (CartPole, MountainCar, Acrobot, LunarLander).

**Strengths:**

1. The paper addresses an important research direction, combining evolution and life-time learning

2.The experiments demonstrate faster and more stable convergence, even in environments that do not require lifetime adaptation. The authors provide reasonable mechanistic explanations (reward smoothing and intrinsic regularization from Oja/BCM rules).

3. The approach could easily be applied to other neuroevolution algorithms beyond NEAT.

**Weaknesses:**

1. Evaluation is restricted to low-dimensional control tasks  that do not require within-lifetime adaptation. This makes it difficult to assess NEOL’s claimed advantage as an “online learning” or “adaptive” system. What about the T-maze or something GoalDirection HalfCheetah?
2. No separate hyperparameter tuning between NEAT and NEOL. Both methods use the same NEAT settings, except for additional plasticity parameters.
3. NEOL is compared only to NEAT and an ablated NEAT (η=0). Missing are comparisons to e.g. Najarro & Risi (2020) and other meta-plasticity or evolutionary meta-learning methods.
4. Limited novelty and missing early work. Other approaches have already combined NEAT with plasticity and are not mentioned, e.g. "Evolving adaptive neural networks with and without adaptive synapses" by Stanley et al.

**Questions:**

1. Have you attempted any environments requiring within-lifetime adaptation (e.g., GoalDirection Cheetah, T-Maze)?
2. How is the approach different to adaptive NEAT by Stanley et al?

---

### Official Review · Reviewer_w2fR · 2025-11-01

**Soundness:** 2
**Presentation:** 3
**Contribution:** 1
**Rating:** 2
**Confidence:** 4

**Summary:**

The authors propose NEOL, a Neuroevolution approach that combines NEAT for evolving topologies of neural network architectures with online learning mechanims implemented via Hebbian learning (or similar). The authors show that this approach surpasses NEAT in performance across several locomotion task in an RL setting.

**Strengths:**

The approach is well motivated and the authors explain it clearly. The experimental setting is sensible with a collection of the popular RL tasks serving as way to compare both approaches.

**Weaknesses:**

First, the claim feels very weak, but more importantly, it has already been done (https://www.cs.utexas.edu/~nn/downloads/papers/stanley.cec03.pdf). The authors need to explain what is new regarding their approach and why this is interesting compared to previous work.

However, even if it was a completely novel approach, it is not clear to me that it is that surprising. Under a Lamarckian setting, if the changes in model weight transfer to offspring during evolution, then is it really that surprising that NEOL surpasses NEAT? That's the minimum I would expect, unless I am misunderstanding something. The authors claim that they include this control but don't show it.

**Questions:**

1. How does this compare to standard RL. Is there a particular setting where this is better? I am happy to also disregard this question if the authors give me some justification (e.g. they wish to explore more biologically plausible models), but they they need to give me a biologically interesting question they wish to answer. Currently there is a biological inspiration, but no particular question they wish to answer.
2. How much of the learning is driven by topological changes? Do models become deeper for example? There is currently no analysis of how the model is driving performance.

---

### Note · Authors · 2025-11-19

**Comment:**

**1.Summary of Reviews**

We sincerely thank reviewers w2fR, XEuE, q1Gg, and bgzr for their detailed and helpful feedback on our NEOL submission.
In general, the reviews see value in the idea of combining evolution and lifetime learning, but they also point out several issues in how we present and evaluate this idea.

**Strengths**
- The paper studies an important and conceptually interesting direction: combining NEAT-style topology evolution with online synaptic plasticity (Hebb, Oja, BCM with reward modulation) in an outer-loop / inner-loop framework. (XEuE, bgzr, w2fR)
- The approach is well motivated and clearly explained. (w2fR)
- The experimental setting uses standard RL benchmarks and is considered sensible. (w2fR)
- The experimental comparison to NEAT is thorough, using many random seeds, several classic control benchmarks, and statistical tests, and it shows faster, more stable, and higher final performance than NEAT. (q1Gg, XEuE)
- The framework can in principle be applied to other neuroevolution algorithms beyond NEAT. (XEuE)

**2.Main Issues Raised by Reviewers**

**Unclear Novelty and Related Work (w2fR, XEuE, q1Gg, bgzr)**

Reviewers ask us to explain more clearly how NEOL differs from earlier work, such as Stanley 2003 (“Evolving adaptive neural networks with and without adaptive synapses”) and later meta-plasticity / evolutionary meta-learning methods. In the current paper, we do not give enough discussion or comparison to highlight our contribution.
Our goal in this work is not to propose a new biological plasticity rule, but to study the hybrid methods for algorithmic design: a reward-modulated Hebb/Oja/BCM rules combined with NEAT in an outer-loop / inner-loop system, and to show how this affects performance, stability, and sample efficiency. We did not state this focus clearly enough.

**Benchmark Setting and Baselines (XEuE, q1Gg, bgzr)**
Reviewers point out that our benchmark setting is limited and partly ambiguous. We mainly use low-dimensional control tasks where lifetime adaptation is not strictly required, and we do not test environments like T-maze or GoalDirection HalfCheetah that would better demonstrate online adaptation.
They also note missing baselines. In particular, we do not compare to standard modern RL methods such as PPO, SAC. This makes it hard to judge the practical value of NEOL beyond being better than NEAT.

**Outer-Loop / Inner-Loop Framework (bgzr, q1Gg)**

 Reviewer bgzr and q1Gg question our use of the outer-loop / inner-loop (meta-learning) framing. In many meta-learning works, the outer loop optimises meta-parameters across a task distribution, and the inner loop adapts to new tasks. In our experiments, we use a single stationary task. We did not explain clearly why we still chose an outer-loop / inner-loop description in this setting, which confuses the purpose of the framework.

**Mechanistic Analysis and Metrics (w2fR, q1Gg, bgzr)**

Reviewers ask for more mechanistic analysis of NEOL. For example, we mention a Lamarckian vs. Darwinian setting (WRITE_BACK=True/False), but we do not report the corresponding ablation results. We also do not analyse how topology evolution (e.g., depth and connectivity) contributes to the observed performance gains.
In addition, our definition of the interaction budget (M or B) and the SCORE metric is incomplete. We do not clearly state whether M is in terms of episodes, steps, or transitions, and we do not explain how we normalise runs with different population sizes. This lack of a clear definition makes it hard for reviewers to interpret the results. These are issues in the current version and need to be fixed.

**Biological Motivation (w2fR, bgzr)**

Reviewers say that our biological motivation is not clear. We use Hebb, Oja, and BCM rules with reward modulation and mention biological plausibility, but we do not specify which biological question or phenomenon we want to explain.
This reflects a mismatch between what we intended and what we wrote: in this paper, our main interest is algorithm design with biologically inspired plasticity, not explaining a specific biological experiment or proposing a new plasticity rule. We need to state this clearly and avoid suggesting that the main goal is a new biological mechanism.

**Clarification on the Contribution of NEOL**

Our paper is to show that a scalable combination of NEAT with fixed, local, reward-modulated plasticity (Hebb/Oja/BCM) in an outer-loop / inner-loop framework can directly improve stability, sample efficiency, and final performance compared to standard NEAT, and to provide a detailed analysis of how evolution and plasticity interact.
The current manuscript does not communicate this point clearly enough and does not include enough comparisons and analysis to fully support it. The reviews make this gap very clear.

**3.Plan for Withdrawal and Future Work**

Given the above issues, we have decided to withdraw the current submission in its present form. A proper revision needs more than adding a few extra tasks or citations. It requires changes to the problem setup, benchmarks, baselines, and analysis, as well as clearer writing.

*We plan to:*

- Clarify novelty and positioning: Rewrite the related work and positioning section to clearly state how NEOL relates to Stanley 2003 and later meta-plasticity / evolutionary meta-learning methods, and to make our algorithmic focus explicit.

- Add more benchmarks and baselines: Include tasks that require within-lifetime adaptation (such as T-maze or GoalDirection HalfCheetah) and add missing modern baselines from RL and neuroevolution.

- Add mechanistic analysis: Provide a clear Lamarckian vs. Darwinian (WRITE_BACK vs. no WRITE_BACK) ablation, analyse how topology evolves, and give precise definitions and usage of M, B, and the SCORE metric.

- Clarify the role of biological plasticity rules: Explain that Hebb/Oja/BCM + reward-modulation are used as fixed, biologically inspired components in an algorithm, and that we do not aim to introduce a new biological plasticity rule or explain a specific biological phenomenon in this paper.

We thank the reviewers again for their valuable insights, which significantly help us improve the work. Once the issues raised by the reviewers are properly addressed, we will resubmit a stronger and clearer manuscript in the near future.

**Withdrawal Confirmation:**

I have read and agree with the venue's withdrawal policy on behalf of myself and my co-authors.